# Synthesis and Third-Order Nonlinear Synergistic Effect of ZrO_2_/RGO Composites

**DOI:** 10.3390/nano11102741

**Published:** 2021-10-16

**Authors:** Xinting Zhao, Fangfang Wang, Jiawen Wu, Baohua Zhu, Yuzong Gu

**Affiliations:** 1Physics Research Center for Two-Dimensional Optoelectronic Materials and Devices, School of Physics and Electronics, Henan University, Kaifeng 475004, China; zxt13164377838@163.com (X.Z.); wujiawen619@163.com (J.W.); 2Key Laboratory of Infrared Imaging Materials and Detectors, Shanghai Institute of Technical Physics, Chinese Academy of Sciences, Shanghai 200083, China; wangfangfang@mail.sitp.ac.cn

**Keywords:** ZrO_2_/RGO nanocomposites, third-order nonlinear optical property, Z-scan technique

## Abstract

Tuning the third-order nonlinear properties of graphene by hybrid method is of great significance in nonlinear optics research. ZrO_2_/reduced graphene oxide (RGO) composites with different ZrO_2_ concentrations were prepared by a simple hydrothermal method. The morphology and structure show that ZrO_2_ nanoparticles were uniformly dispersed on the surface of graphene nanosheets. The nonlinear optical (NLO) characteristics of composites with different ZrO_2_ concentrations were studied by the Z-scan technique of 532 nm picosecond pulsed laser. The results showed that ZrO_2_/RGO composites had saturated absorption and positive nonlinear refraction characteristics. Meanwhile, the third-order nonlinear susceptibility of the ZrO_2_/RGO composite with a 4:1 mass ratio of ZrO_2_ to graphene oxide could reach 23.23 × 10^−12^ esu, which increased tenfold compared to RGO, and the nonlinear modulation depth reached 11.22%. Therefore, the NLO characteristics could be effectively adjusted by controlling the concentration of ZrO_2_, which lays a foundation for further research on the application of ZrO_2_/RGO composites in NLO devices.

## 1. Introduction

Nonlinear optical (NLO) materials are very important for application in optical communication, optical computation and optical signal processing [1,2,3], and are of great importance in the study of materials with excellent nonlinear properties [4,5,6,7]. Graphene is a hexagonal two-dimensional planar crystal material with comprising carbon atoms of sp^2^ hybridized orbitals [8]. Due to its unique photoelectric properties and large specific surface area, graphene has been widely developed in the optical field. The third-order NLO properties of graphene have been proven in some studies [9,10,11]. However, the nonlinear properties of a single graphene material are not sufficient because graphene is a zero-bandgap structure. In order to expand the application of graphene and improve the optical nonlinearity, people have studied graphene hybrid composites, which have obvious nonlinearity [12,13,14]. Therefore, hybrid composites, such as transition metal sulfides and metal selenides, which were compounded with graphene to enhance the nonlinear properties, have been developed [4,15,16]. Rajeswari et al. studied the strong reverse-saturation absorption (RSA) property of WO_3_/reduced graphene oxide (RGO) hybrid material under the excitation of 532 nm, and obtained the third-order nonlinear optical susceptibility χ^(3)^ of 9.727 × 10^−8^ esu [17]. Jiang et al. studied the graphene-titanium dioxide hybrid nanostructure and observed an RSA effect at 700 nm and a saturation absorption (SA) effect at 1100 nm [18]. As a source of Zr in the same family as Ti, zirconia (ZrO_2_) has become an attractive functional material due to its high strength, good thermal stability, excellent optical properties and wide bandgap. It is also non-toxic and has eco-friendly characteristics [19,20,21,22], which is why we chose to study ZrO_2_ in this work.

In this paper, a simple one-step hydrothermal method will be used to synthesize ZrO_2_/RGO nanocomposites including in-situ synthesis of ZrO_2_ nanoparticles and reduced graphene oxide by hydrazine hydrate, and Polyvinyl Pyrrolidone will be added to solve the problem of ZrO_2_ agglomeration during the synthesis of ZrO_2_/RGO nanocomposites. The structure and morphology of the samples were characterized by The X-ray diffraction patterns, scanning electron microscopy, transmission electron microscopy, Fourier transform infrared spectroscopy, UV-Vis absorption spectroscopy and Raman spectroscopy. The properties of nonlinear saturated absorption and positive nonlinear refraction of ZrO_2_/RGO nanocomposites were studied and discussed with regard to the laser wavelength of 532 nm.

## 2. Experimental

### 2.1. Synthesis of ZrO_2_/RGO

Synthesis of graphene oxide (GO): GO was synthesized by a modified Hummer’s method [23]. Firstly, 100 mL of concentrated sulfuric acid and 10 mL of concentrated phosphoric acid were added into a three-necked flask. Secondly, 5 g of potassium permanganate and 0.5 g of carbon powder were added into the three-necked flask which was heated for 24 h in an oil bath at 70 °C. The reactants were cooled to room temperature after the reaction was finished, and then were poured into a 150 mL ice mixture of H_2_O_2_ and distilled water, which was stirred to room temperature by using a glass rod. Finally, the mixture was washed once by using hydrochloric acid, four times by using ultra-pure water and was freeze-dried for 24 h.

Synthesis of the ZrO_2_/RGO: Firstly, 10 mg of GO was placed in 40 mL of ultra-pure water and was sonicated for 2 h. 100 mg ZrOCl_2_·8H_2_O and 10 mg polyvinyl pyrrolidone were added to GO solution and stirred until they were fully mixed. 200 μL N_2_H_4_·H_2_O was slowly added to the solution and stirred for 20 min. Secondly, the mixture was placed in a 50 mL autoclave, sealed, and subjected to hydrothermal reaction at 180 °C for 12 h. The black product was centrifuged and washed repeatedly with ultrapure water. Finally, it was dried in vacuum at 60 °C for 24 h. In this study, the mass ratios of GO and ZrO_2_ were 1:2, 1:4, 1:6, and 1:8, respectively, which were obtained by changing the dosage of ZrO_2_ and selenium powder, and were labeled as T1, T2, T3, and T4, respectively. Pure ZrO_2_ nanocrystals and RGO were synthesized by the same process. The fabrication schematic of the ZrO_2_/RGO nanocomposites is shown in Figure 1.

### 2.2. Characterization

The X-ray diffraction (XRD) patterns were measured using a DX-2500 diffractometer. The morphology and structure of the samples were characterized using field emission scanning electron microscopy (SEM, Carl Zeiss Inc., Oberkochen, Baden-Württemberg, Germany) and transmission electron microscopy (TEM, JEOL JEM-2100 operating at 200 kV, JEOL Ltd. Inc., Akishima, Tokyo, Japan). The Fourier transform infrared (FTIR) spectra were taken using a FTIR VERTEX 70 v (Bruker Optics Corp. Rudolph Planck Stetter 2776275, Ettlingen, Germany). The UV-Vis absorption spectra of samples were measured using a Perkin-Elmer Lambda 35 spectrometer. The Raman spectrum was observed and recorded on a Lei Nishao-based Raman spectrometer with an excitation wavelength of 532 nm. The third-order NLO characteristics were measured by Z-scan technology with laser source of Nd: YAG laser system (EKSPLA, PL2251. Savanoriu Ave 237 LT, Vilnius, Lithuania), wavelength of 532 nm, pulse width of 30 ps and pulse repetition frequency of 10 Hz. Before measurement, the optical path was tested and calibrated by using standard nonlinear material CS_2_.

## 3. Results and Discussion

The morphology of the samples GO, ZrO_2_ and ZrO_2_/RGO nanocomposites was characterized by scanning electron microscopy and transmission electron microscopy. The SEM images of the samples (Figure 2a–f) showed that graphene was multi-layer, and the zirconia monomer presented spherical nanoparticles. And after the reaction between Zr^+^ and GO, zirconia was attached to the graphene. When GO: ZrO_2_ = 1:2, the dispersibility of the particles on the graphene was relatively good. With the increase of the concentration of ZrO_2_, the zirconia on the graphene wa gradually increased. Figure 2g,I are the TEM images of T2 and T4, and the size distribution of ZrO_2_. The nearly circular ZrO_2_ crystal is attached to the graphene, and the average size of ZrO_2_ was about 7.4 nm. The results were consistent with SEM.

The structures of the samples GO, ZrO_2_, and ZrO_2_/RGO nanocomposites were analyzed by X-ray diffraction, as shown in Figure 3. GO had a major diffraction peak at 2θ = 10.44°, which corresponded to the (002) plane of GO [24]. The diffraction peak of GO disappeared in the composite, suggesting that GO was transformed to G. The diffraction peaks of ZrO_2_ were at 2θ = 30.12°, 34.92°, 50.21°, 59.67°, and 60.12°, which were corresponded to (111), (200), (220), (311) and (211) crystal faces of ZrO_2_, respectively, and were consistent with that of the standard colorimetric card (JCPDS no.89-9069). The diffraction peaks of composites were in accord with that of ZrO_2_, indicating that ZrO_2_ was successfully embedded on RGO. The size of ZrO_2_ nanoparticles on RGO were obtained by the Debye-Scheler equation [25]: D = kλ/βcosθ. D is the particle size, *k* is a dimensionless shape factor, λ is wavelength of incident radiation, β is the line broadening in radians at half maximum intensity, θ is the Bragg angle. The calculate crystal average particle size of the composites was close to the results obtained in TEM characterization.

FT-IR spectrum provided the functional group analysis of GO, ZrO_2_, and ZrO_2_/RGO nanocomposites. As shown in Figure 4, the absorption peaks of GO appeared in 3394 cm^−1^, 2973 cm^−1^, 1650 cm^−1^, and 1048 cm^−1^ in the FT-IR spectrum, which respectively correspond to O-H vibration of hydroxyl group, C-H vibration of saturated CH_2_, C=C vibration of aromatic hydrocarbon and C-O vibration of aromatic hydrocarbon in graphene [26]. In the FT-IR spectrum of the ZrO_2_/RGO composite, with the decrease of the concentration of ZrO_2_, the strength of oxygen-containing functional groups in GO hydroxyl and carboxyl groups were reduced or even disappeared. The vibration peaks at 503 cm^−1^, 592 cm^−1^, and 606 cm^−1^ of ZrO_2_/RGO nanocomposites would be attributed to the vibration of the Zr-O functional group [27]. This indicated that ZrO_2_ nanoparticles were successfully embedded on RGO.

Raman analysis is a very important mean to characterize the sp^3^ and sp^2^ hybridized carbon atoms in graphene [28]. As shown in Figure 5, under the excitation at the wavelength of 532 nm, the composites showed two characteristic peaks at 1593 cm^−1^ and 1335 cm^−1^, which corresponded to the G and D bands of graphene, respectively. The D-band was related to the defect state of graphene, and the G-band came from the in-plane vibration of carbon atoms in graphene [29]. The intensity ratios of D to G for GO and composites T1–T4 were 0.76, 0.83, 0.850, 0.853, and 0.86, respectively. The intensity of D-band/the intensity of G-band (I_D_/I_G_) of the composites increased, indicating that the surface defects of the composites increased with the increase of ZrO_2_ concentration.

Figure 6 shows the UV-Vis absorption peak position of the sample. The absorption peak of GO was at 228.96 nm, which might be mainly derived from the π-π* transition of C-C and C=C in sp^2^ hybridization [30]. For ZrO_2_/RGO nanocomposites, with the increase of ZrO_2_ concentration, the absorption peaks were located at 262.79 nm, 264.16 nm, 266.07 nm, and 275.15 nm, respectively. The absorption peak underwent red shift. The energy gap (E_g_) of ZrO_2_ was estimated to be 5.47 eV, and the band gaps of ZrO_2_/RGO composites T1–T4 were calculated from UV absorption peaks as 4.72 eV, 4.69 eV, 4.66 eV and 4.51 eV, respectively. The red shift of the absorption peak and the reduction of the band gap provided the basis for the covalent bond between graphene and ZrO_2_. The high refractive index of ZrO_2_ and the high electron mobility of graphene, as well as the charge transfer between graphene and ZrO_2_, made the band gap of the composite become smaller.

### Nonlinear Optical Effect of ZrO_2_/RGO

The samples were dissolved in anhydrous ethanol at 0.2 mg/mL, and the input pulse intensity was 7 GW/cm^2^. The open-aperture (OA) and closed-aperture/open-aperture (CA/OA) Z-scan curve of samples were shown in Figure 7, where the scattered point represented the experimental transmittance and the solid line represented the theoretical fitting. Anhydrous ethanol had no peak under the same conditions, so its influence could be negligible. Figure 7a was the OA Z-scan curve of RGO, ZrO_2_ and composite T2. RGO, ZrO_2_ and composite T2 exhibited nonlinear saturated absorption at the focus, and the saturated absorption of composite T2 was stronger than that of RGO and ZrO_2_. Since RGO had a zero-band-gap structure, it would theoretically absorb any wavelength. When the strong laser irradiated RGO, the electrons in the valence band absorbed the energy of the photon and continuously excited to the conduction band, and the photon energy subband of the valence band and the conduction band were completely occupied by the electrons and holes, resulting in the blocked transition, thus achieving saturated absorption. The main reason for the SA of ZrO_2_ was that the electrons in the valence band absorbed the photon energy and then stimulated the conduction band when the laser irradiated the ZrO_2_. The essence of this excitation was that the electrons in the 2p energy state of oxygen atom in the valence band excited to the 4d energy state of zirconium nanoparticle. According to Fermi-Dirac distribution [15,31,32], the energy of electron absorption photons was excited from a low level to a high level, and the excitation rate of absorption transition was much higher than the relaxation rate of carriers. The photonic bands in the valence and conduction bands were completely occupied by electrons and holes, resulting in hindered transitions and saturated absorption. Figure 7b was the OA Z-scan of composites T1–T4, which showed SA. With the increase of ZrO_2_ concentration, the saturated absorption increased firstly and then decreased, which indicated that the combination of ZrO_2_ and RGO played a synergistic effect on the nonlinear property. In order to better explain the SA of composites, theoretical fitting was adopted, and the formula combining the SA and RSA coefficient was used [33,34,35]:$$\alpha \left(I\right)=\alpha_{0}\left(1+I/I_{s}\right)+\delta I$$

α(I) represented the total absorption of the sample, $\alpha_{0}$ was the linear absorption coefficient which could be obtained from the UV-vis spectrum. $I_{s}$ was the SA intensity, and *δ* was the RSA coefficient. The experimental data of OA Z–scan were fitted with the equation below [33,35]:
$$T(z)=\sum_{m=0}^{\infty }\frac{{\left(-\beta I_{0}L_{eff}\right)}^{m}}{{\left(1+\frac{Z^{2}}{Z_{R}^{2}}\right)}^{m}{\left(1+m\right)}^{\frac{3}{2}}}$$
where T(z) was the OA Z-scan normalized transmittance, z was the sample position, *I*_0_ was the laser intensity at the focus, z_0_ = $\pi \omega_{0}^{2}/\lambda$ was the length of Rayleigh diffraction, and ω_0_ was the laser beam waist. *L_eff_* = [1 − exp($\alpha_{0}$L)]/$\alpha_{0}$, was the effective thickness of the sample, L was the actual thickness of the sample. The effective nonlinear absorption coefficient of the sample could be obtained by fitting the curve: *β* = 2^3/2^(1 − T_Z=0_)(1 + z^2^/z_0_^2^)/*I*_0_*L_eff_*, The imaginary part of the third order nonlinear susceptibility of the sample was calculated by Im χ(3)=cn02λβ/480π3.

Figure 7c,d shows the CA/OA track of RGO, ZrO_2_, and composite samples T1–T4. It was shown that the RGO, ZrO_2_ and composites had self-focusing properties. With the increase of ZrO_2_ concentration, the nonlinear refraction firstly decreased and then increased, which indicated that the combination of ZrO_2_ and RGO had a synergistic effect on the nonlinearity. The normalized CA/OA Z-scan experimental data were fitted by the following equation [30]:$$T(z)=1-\frac{4x\Delta \emptyset_{0}}{\left(x^{2}+9\right)\left(x^{2}+1\right)}$$
where x = z/z_0_, $\Delta \emptyset_{0}$ was the on-axis phase shift at the focus, defined as $\Delta \emptyset_{0}=k\Delta n_{0}L_{\mathrm{eff}}={k\gamma I}_{0}L_{\mathrm{eff}}$, k = 2π/λ was the wave vector, λ was the laser wavelength, $\gamma =\lambda \alpha \Delta T_{P-V}/\left(0.812{\pi I}_{0}{\left(1-S\right)}^{0.25}\left(1-e^{-\alpha L}\right)\right)$, γ was the nonlinear refractive index coefficient in m^2^/W, $\Delta T_{P-V}$ represented the difference between the normalized transmitted peaks and valleys of the open/closed Z-scan trace, S was the fluence of the aperture. Then, the nonlinear refraction index was calculated by $n_{2}\left(\mathrm{esu}\right)=\left({\mathrm{cn}}_{0}/40\pi \right)\gamma \left(m^{2}/W\right)$. Thus, the real part of the third order nonlinear susceptibility was calculated using Reχ (3)=n0n2/3π, and the third-order nonlinear susceptibility was obtained by χ(3)=[(Re χ(3))2+(Im χ(3))2]12. The results of I_s_, Im χ ^(3)^, Re χ ^(3)^, χ ^(3)^ and ΔT for all the samples were presented in Table 1.

From the table, one can see that the Imχ^(3)^ of the composites firstly increased and then decreased with the increase of ZrO_2_ concentration, and the Imχ^(3)^ of T2 reached the maximum whose value was 23.19 × 10^−12^ esu. In addition, the real part of the NLO susceptibility Reχ^(3)^ of the composites increased, and Reχ^(3)^ of T4 reached the maximum of 1.74 × 10^−13^ esu. The third-order NLO susceptibility χ^(3)^ was also tuned and enhanced, and the maximum χ^(3)^ value of T2 is 23.23 × 10^−12^ esu, which was approximately 10 times that of RGO and 20 times that of ZrO_2_. These data indicated that the NLO properties of the ZrO_2_/RGO composites were obviously better than those of the ZrO_2_ nanoparticles and RGO nanosheets.

## 4. Discussion

As shown in Figure 7 and Table 1, the nonlinear absorption and refraction characteristics of RGO were regulated by ZrO_2_. Particularly with the increase of ZrO_2_ concentration, the nonlinear saturation absorption first increased and then decreased, which might result from the charge transfer effect between ZrO_2_ and RGO [36]. To explain the regulation of charge transfer effect on the NLO properties of ZrO_2_/RGO composites, a model of charge transfer was proposed based on the valence band (VB) spectrum and relative energy levels as given in Figure 8 [30,31,36]. ZrO_2_ is a wide bandgap material (the bandgap is 5.47 eV) and RGO is zero bandgap. Therefore, the two materials could be regarded as a donor-acceptor system. Under the excitation of 2.33 eV (532 nm), the 2p energy state of oxygen atom in the valence band of ZrO_2_ underwent spin splitting and bonding/anti-bonding splitting to generate electrons and oxygen holes, resulting in an electron-guiding band transition, which was transformed from an excited state to a more intense state [19,37]. Meanwhile, the electrons in graphene absorbed the photon energy and transition from the ground state to the excited state. The electrons generated in the ZrO_2_ valence band would not only be relaxed to the conduction band through oxygen vacancy transition, but also be transferred to the excited state of RGO through charge transfer and then returned to the ground state. Therefore, the charge transfer process destroys the electron relaxation in RGO, hinders the linear absorption of graphene, and increases the saturated absorption of graphene.

In order to better understand the electron relaxation in the SA of ZrO_2_/RGO composites, the modulation depth, ΔT, defined as the difference between the maximum and minimum transmissions, ΔT = $T_{\max}-T_{0}$ was introduced, which was calculated by the Franz-Norvik equation [38,39]:$$T\left(F\right)=T_{0}+\frac{T_{\mathrm{FN}}-T_{0}}{1-T_{0}}\left(T_{\max}-T_{0}\right)$$
where $T_{0}=e^{-\sigma_{g}\mathrm{NL}}$ was the linear transmission, $T_{\max}=e^{-\sigma_{e}\mathrm{NL}}$ was the saturable transmission, $\sigma_{g}$ and $\sigma_{e}$ represented the absorptive cross sections of ground and excited states, respectively. N represented the density of the absorptive centers (density of atoms for approximation), and L was the sample thickness. As shown in Table 1, the modulation depths of composites T1–T4 were 9.62%, 11.22%, 8.95% and 10.98%, respectively, which were larger than that of monomer graphene. In addition, the saturated absorption intensity firstly decreased and then increased, and the minimum value I_s_ was 0.58 GW/cm^2^, appearing at composite sample T2. It might arise from the electron relaxation and charge transfer processes in ZrO_2_/RGO composite. The electron relaxation and charge transfer processes were related to the bandgap of material. ZrO_2_ is a wide bandgap material, and the excitation of electrons from the valence band to the conduction band required a relatively large amount of energy. The size of ZrO_2_ in ZrO_2_/RGO composite became smaller (7.4 nm), which was obtained by SEM, TEM and XRD characterization. The small size effect made the change of electronic relaxation between the ZrO_2_ and graphene. Therefore, under the excitation of strong light, the electrons generated from the valence band of ZrO_2_ in ZrO_2_/RGO composites relaxed to the conduction band through the oxygen vacancy transition. When the concentration of ZrO_2_ increased, the band gap of the composite became smaller and the oxygen vacancy increased. This resulted in few electrons being transferred to RGO, and the relaxation damage to electrons in RGO became smaller, and the SA of the composite was reduced. the modulation depth of ZrO_2_/RGO composite with the mass ratio of 4:1 reached the maximum value of 11.22%, indicating that charge transfer between RGO and ZrO_2_ made the longer electron relaxation time, which hindered the linear absorption of graphene and increased the saturated absorption of the composite.

Compared with the reported materials, the χ^(3)^ of composite T2 (23.23 × 10^−12^ esu) was three times larger than that of graphene–γMnS composite (6.23 × 10^−12^ esu) [40], four times larger than that of α–MnS/rGO composite (4.93 × 10^−12^ esu) [41], and six times larger than that of the rGO–Au@CdS composite (3.43 × 10^−12^ esu) [31]. Meanwhile, χ^(3)^ of composite T2 was twenty times larger than that of GeS_2_–Ga_2_S_3_–CDS chalcogenide glass (0.165 × 10^−12^ esu) [42], and six orders of magnitude larger than that of CdS quantum dots (3.96 × 10^−18^ esu) [43]. Therefore, the ZrO_2_/RGO composite exhibited relatively strong NLO properties.

## 5. Conclusions

ZrO_2_/RGO nanocomposites were synthesized by the hydrothermal method and their NLO properties were studied by the Z-scan technique. The experimental results showed that ZrO_2_/RGO composites had nonlinear saturated absorption and positive nonlinear refraction properties. By controlling the content of ZrO_2_, the third-order NLO susceptibility χ^(3)^ of the composite first increased and then decreased, and the maximum value was 23.23 × 10^−12^ esu, which was about 10 times that of RGO, and the modulation depth of the composite reached 11.22%. The enhanced NLO property of ZrO_2_/RGO composites indicates its possible application in many NLO devices such as optical communication, optical switches and optical storage.

## Figures and Tables

**Figure 1 nanomaterials-11-02741-f001:**
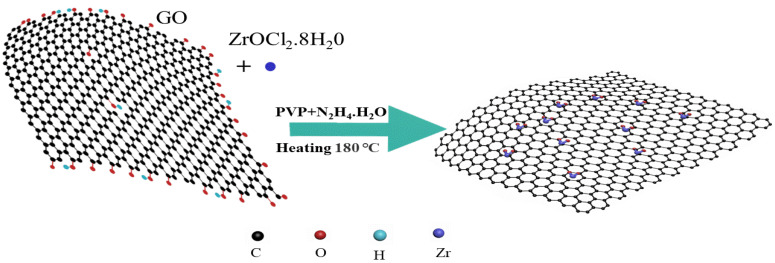
Schematic diagram of the ZrO_2_/RGO composite nanostructure.

**Figure 2 nanomaterials-11-02741-f002:**
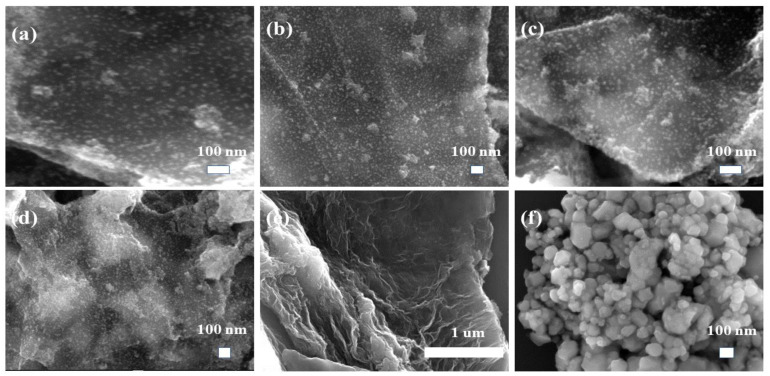
(**a**–**f**) SEM images of ZrO_2_/RGO composite T1–T4, RGO, ZrO_2_. (**g**,**h**) TEM images of composite T2 and T4. (**I**) the size distribution of ZrO_2_.

**Figure 3 nanomaterials-11-02741-f003:**
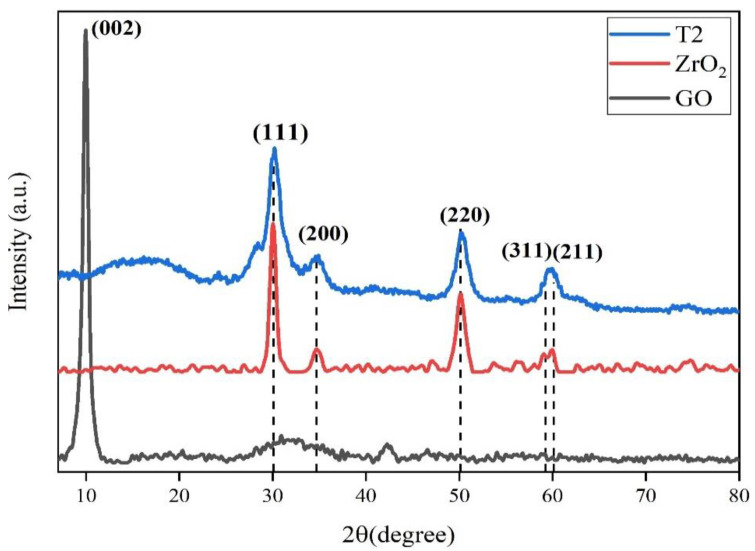
XRD pattern of RGO, ZrO_2_, and ZrO_2_/RGO composite T2.

**Figure 4 nanomaterials-11-02741-f004:**
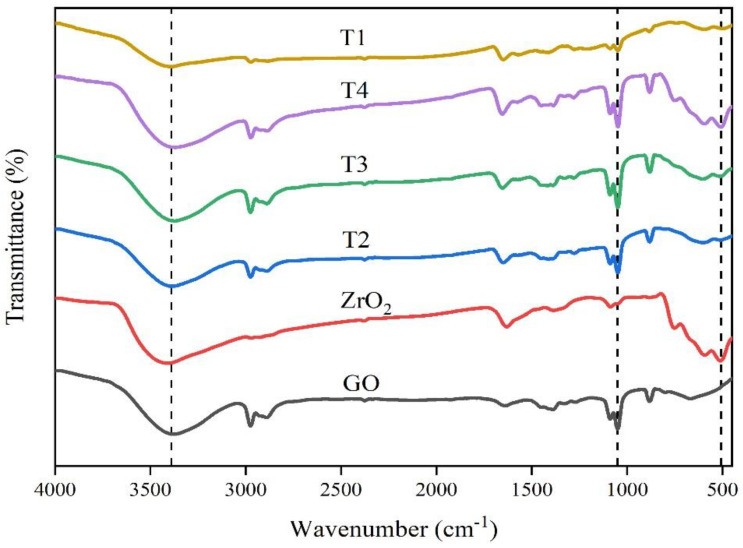
FT-IR spectra of GO, ZrO_2_, and ZrO_2_/RGO composite T1–T4.

**Figure 5 nanomaterials-11-02741-f005:**
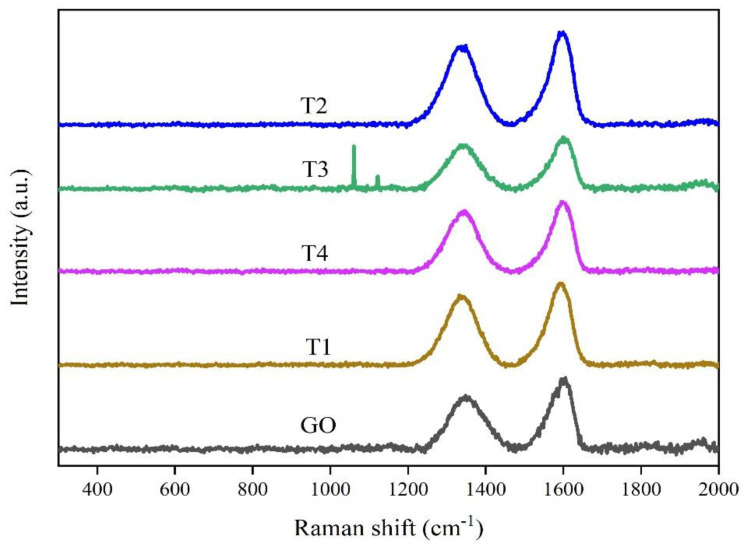
Raman patterns of GO and ZrO_2_/RGO composites T1–T4.

**Figure 6 nanomaterials-11-02741-f006:**
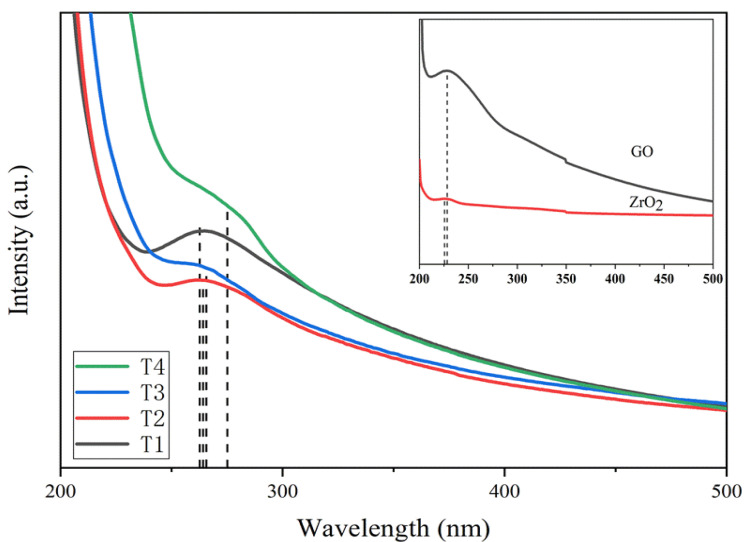
UV-Vis absorption spectra of ZrO_2_ and the ZrO_2_/RGO composites T1–T4.

**Figure 7 nanomaterials-11-02741-f007:**
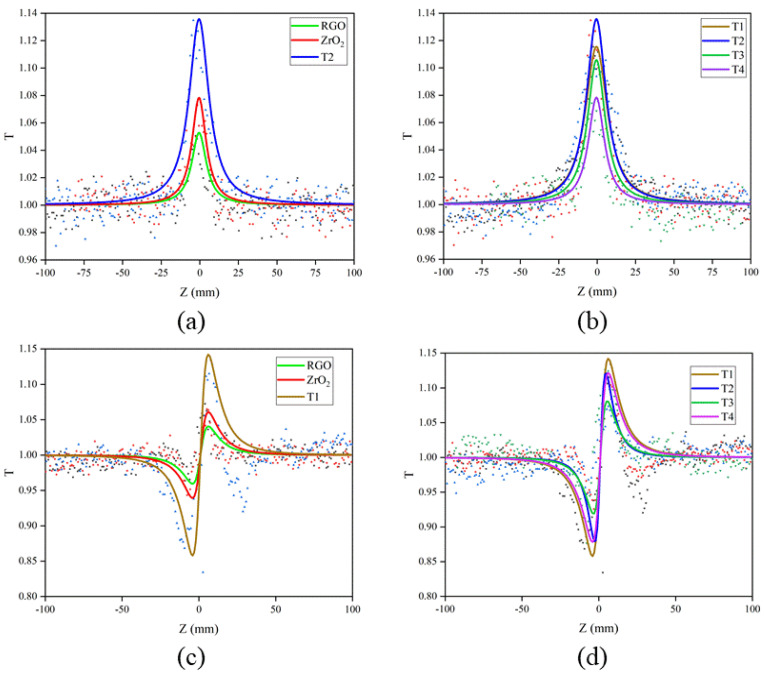
(**a**) Open-aperture Z-scan curves of RGO, ZrO_2_, and the ZrO_2_/RGO composite T2; (**b**) Open-aperture Z-scan curves of composites T1–T4; (**c**) Closed-aperture/open–aperture Z-scan curves of RGO, ZrO_2_, and the ZrO_2_/RGO composite T2; (**d**) Closed-aperture/open-aperture Z–scan curves of composites T1–T4.

**Figure 8 nanomaterials-11-02741-f008:**
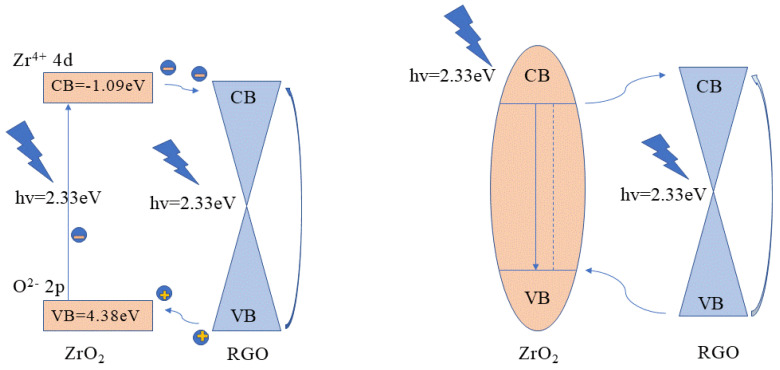
The charge transfer mechanism and energy-level diagram of the ZrO_2_/RGO composites.

**Table 1 nanomaterials-11-02741-t001:** The nonlinear susceptibilities of the samples.

	I_s_	Imχ ^(3)^	Reχ ^(3)^	χ ^(3)^	ΔT
Sample	GW/cm^2^	/10^−12^ esu	/10^−13^ esu	/10^−12^ esu	/%
RGO	9.60	−2.31	0.63	2.39	4.97
ZrO_2_	1.64	−1.79	0.18	1.81	7.73
T1	1.52	−8.413	1.06	8.48	9.62
T2	0.58	−23.19	1.49	23.23	11.22
T3	3.32	−6.82	1.65	6.88	8.95
T4	4.91	−4.89	1.74	5.19	10.38

## Data Availability

Data provided in this study are available upon request from the corresponding authors. These data cannot be made public due to the applied experimental studies being to be conducted.

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
