# Peer review of "Synthesis and Third-Order Nonlinear Synergistic Effect of ZrO2/RGO Composites"

_nanomaterials, 2021, doi:10.3390/nano11102741_

Round 1
Reviewer 1 Report
The authors embedded ZrO2 nanoparticle into graphene oxide at different ZrO2 concentrations. The produced composites were characterized with several methods, like x-ray diffraction, electron microscopes, Fourier transform, absorption, and Raman spectroscopies to prove they got what they aimed to. The third order susceptibility of the composites were measured with Z-scan technique and concluded that the mass ration of 1:4 resulted the highest susceptibility what was about 10-times of the nonlinear susceptibility of both the ZrO2 nanoparticle and the graphene. An explanatory model of the increased nonlinearity was also established.
Comments:
- The sample preparation and characterization are well written and understandable. However, the explanation especially in lines between 240 and 262 are difficult to read and follow.
- Line 73: “mass ratios of GO and ZrO 2 were 1:2, 1:4, …” instead of “mass ratios of GO and ZrO 2 of 1:2, 1:4, …”
- Line 96 and Figure 2: it is written “it could be clearly seen from the image of the ZrO2/RGO composite that zirconia was uniformly distributed on the graphene”. I have not daily, only some routines with these techniques, but I cannot see this. Also, I cannot recognize the graphene and also the nanoparticles in panels (g) and (h).
- Line 110: “nanoparticles on RGO were made using …”.
- Line 141: It is very hard to see these absorption peaks in Fig. 6 and seems no information at longer than 350 nm.
- Line 169: It would be better to use “ZrO2 nanoparticle” instead of “ZrO2 atom”.
- Lines 184, 185 and 251: It would be better to use simply “saturated absorption intensity”, “anti-saturation absorption coefficient” without the “related to …” parts.
Author Response
Name of journal: nanomaterials
Manuscript Number: nanomaterials-1370353
We sincerely appreciate the editor and all reviewers for their constructive remarks and valuable suggestions, which have significantly raised the quality of the manuscript. Each suggested comment brought forward by the reviewers has been accurately considered. In this revision we have tried to answer the questions and reply to the referees’ comments, point by point. We have highlighted the changes within the revised manuscript in red.
Reviewer #1:
The authors embedded ZrO2 nanoparticle into graphene oxide at different ZrO2 concentrations. The produced composites were characterized with several methods, like x-ray diffraction, electron microscopes, Fourier transform, absorption, and Raman spectroscopies to prove they got what they aimed to. The third order susceptibility of the composites were measured with Z-scan technique and concluded that the mass ration of 1:4 resulted the highest susceptibility what was about 10-times of the nonlinear susceptibility of both the ZrO2 nanoparticle and the graphene. An explanatory model of the increased nonlinearity was also established.
Comments:
- - Line 73: “mass ratios of GO and ZrO2 were 1:2, 1:4, …” instead of “mass ratios of GO and ZrO2 of 1:2, 1:4, …” - Line 110: “nanoparticles on RGO were made using …” - Line 169: It would be better to use “ZrO2 nanoparticle” instead of “ZrO2 atom”.
Response: Thank you so much for your careful check. The errors have been corrected in the revised manuscript.
- The sample preparation and characterization are well written and understandable. However, the explanation especially in lines between 240 and 262 are difficult to read and follow.
Response: Thank you for your useful reminding. The explanation in lines between 240 and 262 have been revised in the revised manuscript.
- - Line 96 and Figure 2: it is written “it could be clearly seen from the image of the ZrO2/RGO composite that zirconia was uniformly distributed on the graphene”. I have not daily, only some routines with these techniques, but I cannot see this. Also, I cannot recognize the graphene and also the nanoparticles in panels (g) and (h).
Response: The scales of SEM and TEM images might be so large that the samples don’t look very clear. The scales of SEM (a)-(d) and TEM images (g) and (h) in Fig. 2 are adjusted to clearly display nanoparticles and graphene.
- - Line 141: It is very hard to see these absorption peaks in Fig. 6 and seems no information at longer than 350 nm.
Response: The absorption peak of GO was at 228.96 nm, and this absorption peak was mainly derived from the π–π transition of C-C and C=C in sp2 hybridization. After the attachment of ZrO2, the absorption peak is redshifted to 262.79 nm. And with the increase of ZrO2 concentration, the peak was redshifted to 275.15 nm. There were no obvious absorption peaks at longer than 350 nm. So the bandgap of the samples were determined by the Tauc relationship: αhν= A(hν-Eg)n, where α is the absorption coefficient, hν is the photon energy, and n=1/2 for the direct bandgap material ZrO2. To state the absorption difference of the samples, we have calculated the bandgap, which is 3.38 eV, 3.16 eV, 3.25 eV, and 3.60 eV for T1, T2, T3, and T4, respectively.
- - Lines 184, 185 and 251: It would be better to use simply “saturated absorption intensity”, “anti-saturation absorption coefficient” without the “related to …” parts.
Response: Thank you for asking this detailed question. We have made changes in the corresponding position of the article.
Please see the attachment.

Reviewer 2 Report
In this manuscript, the authors reported the synthesis of ZrO2 reduced graphene oxide composites (ZrO2/RGO) with varying ZrO2 concentrations, which were characterized by SEM, TEM, X-ray diffraction, FT-IR, Raman, UV-vis measurements, etc. The authors also explored the saturated absorption of ZrO2/RGO with different ZrO2 concentrations by Z-scan technique, where the largest third-order non-linear optical susceptibility of ZrO2/RGO was observed in the mass ratio of GO and ZrO2 of 1:4. The reviewer finds several previous publications on the synthesis and characterization of the ZrO2/RGO (RSC Adv. 2017, 7, 12690-12703; RSC Adv. 2019, 9, 30439-30447, etc.), but this manuscript is the first to report the third-order non-linear optical properties of the ZrO2/RGO with respect to the ZrO2 composition. However, the reviewer finds that this manuscript is overall poorly organized and lacks significant experimental evidence for the charge transfer mechanism responsible for the enhancements in the saturated absorption properties of the ZrO2/RGO. More detailed comments and questions are shown below:
- In this manuscript, there are too many grammatical and typographical errors. It needs extensive language editing before the resubmission.
- The average particle size of the ZrO2/RGO sample was determined as 7.46 nm. But considering the maximum resolution (0.19 nm) of the TEM images by JEOL-2100, the correct value with appropriate significant numbers should be given.
- The assignments for the diffraction patterns of ZrO2/RGO (labeled as T2 in Fig. 3) are inconsistent with the previous reports (RSC Adv., 2019, 9, 30439-30447). Re-assignments for the diffraction patterns or suggestions for the correction of the previous results should be made.
- The authors reported that the absorption maximum wavelength of the ZrO2/RGO red-shifts with increasing the ZrO2 However, in Fig. 6(a), the absorption maximum of the T2 spectrum (a mass ratio of ZrO2 : RGO = 4:1) is slightly red-shifted from that of T3 (a mass ratio of ZrO2 : RGO = 6:1).
- The authors discussed the increases of saturated absorption of the ZrO2/RGO based on the charge transfer mechanism between the electron donor and acceptor of ZrO2 and GO, respectively. However, further experimental evidence or references are needed for this suggestion. Time-resolved spectroscopy measurements such as femtosecond transient absorption can be used to observed the charge transfer in the ZrO2/RGO samples.
Author Response
Responses to Reviewers
Name of journal: nanomaterials
Manuscript Number: nanomaterials-1370353
We sincerely appreciate the editor and all reviewers for their constructive remarks and valuable suggestions, which have significantly raised the quality of the manuscript. Each suggested comment brought forward by the reviewers has been accurately considered. In this revision we have tried to answer the questions and reply to the referees’ comments, point by point. We have highlighted the changes within the revised manuscript in red.
Reviewer #2:
In this manuscript, the authors reported the synthesis of ZrO2 reduced graphene oxide composites (ZrO2/RGO) with varying ZrO2 concentrations, which were characterized by SEM, TEM, X-ray diffraction, FT-IR, Raman, UV-vis measurements, etc. The authors also explored the saturated absorption of ZrO2/RGO with different ZrO2 concentrations by Z-scan technique, where the largest third-order non-linear optical susceptibility of ZrO2/RGO was observed in the mass ratio of GO and ZrO2 of 1:4. The reviewer finds several previous publications on the synthesis and characterization of the ZrO2/RGO (RSC Adv. 2017, 7, 12690-12703; RSC Adv. 2019, 9, 30439-30447, etc.), but this manuscript is the first to report the third-order non-linear optical properties of the ZrO2/RGO with respect to the ZrO2 composition. However, the reviewer finds that this manuscript is overall poorly organized and lacks significant experimental evidence for the charge transfer mechanism responsible for the enhancements in the saturated absorption properties of the ZrO2/RGO. More detailed comments and questions are shown below:
Comments:
- In this manuscript, there are too many grammatical and typographical errors. It needs extensive language editing before the resubmission.
Response: We are sorry for the grammatical and typographical errors. We have carefully revised them in the article.
- The average particle size of the ZrO2/RGO sample was determined as 7.46 nm. But considering the maximum resolution (0.19 nm) of the TEM images by JEOL-2100, the correct value with appropriate significant numbers should be given.
Response: The average particle size of the ZrO2/RGO sample was counted by the TEM images. To make the proof of the average particle size of ZrO2 clear, we added the size distribution in Fig. 2.
- The assignments for the diffraction patterns of ZrO2/RGO (labeled as T2 in Fig. 3) are inconsistent with the previous reports (RSC Adv., 2019, 9, 30439-30447). Re-assignments for the diffraction patterns or suggestions for the correction of the previous results should be made.
Response: We have compared the diffraction patterns of ZrO2/RGO in our work with the previous report (RSC Adv., 2019, 9, 30439-30447). To state more clearly, the characteristic diffraction peaks of (111), (200), (220), (311) and (211) have been added in the Fig. 3. To ensure the XRD data, we have carefully examined the data. Yet, the XRD peaks were not in complete accord with the previous report. Such differences may include the following:
(1). The reaction temperature is different. The reaction temperature is 180 °C in our experiment, and reaction temperature of this report (RSC Adv., 2019, 9, 30439-30447) is 170 °C.
(2). The PH value during the reaction is different. the PH value of the solution is 8 in our experiment, while that of this report solution is 9.
(3). The selected XRD standard card might be different. The selected XRD standard card in our experiment is JCPDS no.89-9069, yet standard card of this report is not mentioned.
- The authors reported that the absorption maximum wavelength of the ZrO2/RGO red-shifts with increasing the ZrO2. However, in Fig. 6(a), the absorption maximum of the T2 spectrum (a mass ratio of ZrO2: RGO = 4:1) is slightly red-shifted from that of T3 (a mass ratio of ZrO2: RGO = 6:1).
Response: In order to display the absorption peak more clearly, the abscissa of Fig. 6 has been shortened. In the revised manuscript, the abscissa wavelength of Figure 6 has been modified to the range of 200 nm to 400 nm. The absorption maximum of T3 (a mass ratio of ZrO2 : RGO = 6:1) was slightly red-shifted from that of T2 (a mass ratio of ZrO2 : RGO = 4:1). To state the absorption difference of the samples, we have calculated the bandgap, which is 3.38 eV, 3.16 eV, 3.25 eV, and 3.60 eV for T1, T2, T3, and T4, respectively.
- The authors discussed the increases of saturated absorption of the ZrO2/RGO based on the charge transfer mechanism between the electron donor and acceptor of ZrO2and GO, respectively. However, further experimental evidence or references are needed for this suggestion. Time-resolved spectroscopy measurements such as femtosecond transient absorption can be used to observed the charge transfer in the ZrO2/RGO samples.
Response: Thank you for the helpful suggestions. Experiment of femtosecond transient absorption is a useful tool to prove the charge transfer process in the composite samples. We have also done this work in our previous published articles to confirm the charge transfer process in composites (L. Li, F. Wang, Y. Liu, F. Cao, B. Zhu, Y. Gu, Local-Field-Dependent Nonlinear Optical Absorption of Black Phosphorus Nanoflakes Hybridized by Silver Nanoparticles, The Journal of Physical Chemistry C 125 (2021) 15448-15457). This paper is added in the revised manuscript to enhance persuasiveness. However, to deeply study the charge transfer between semiconductor particle and RGO, we are carrying out a series of time-resolved spectroscopy measurements and doing theoretical simulation by FDTD calculation. These results will be presented in our future articles in detail.
Please see the attachment.

Reviewer 3 Report
This manuscript presents the synthesis and third-order nonlinear synergistic effect of ZrO2/RGO composites. There are a few remarks that, I hope, can help the authors to improve the text:
- The language should be improved.
- There are some typo errors in the text.
- Abbreviations should be explained in the text
- Authors wrote “absorption peak was mainly derived from the π–π transition”. Are they sure that there is π–π transition?
- Fig. 6 should be corrected.
- It is required to compare the nonlinear values ​​of the studied materials with similar ones.
- Authors should read and cite the following article: Materials 13, 2559 (2020).
- More interpretation of the results should be added.
Summarize, this review article is interesting, and it could be published in Nanomaterials but after major revision.
Author Response
Responses to Reviewers
Name of journal: nanomaterials
Manuscript Number: nanomaterials-1370353
We sincerely appreciate the editor and all reviewers for their constructive remarks and valuable suggestions, which have significantly raised the quality of the manuscript. Each suggested comment brought forward by the reviewers has been accurately considered. In this revision we have tried to answer the questions and reply to the referees’ comments, point by point. We have highlighted the changes within the revised manuscript in red.
Reviewer #3:
This manuscript presents the synthesis and third-order nonlinear synergistic effect of ZrO2/RGO composites. There are a few remarks that, I hope, can help the authors to improve the text:
Comments:
- The language should be improved.
Response: Thanks for your suggestion. The language has been improved in the revised text.
- There are some typo errors in the text.
Response: Thank you so much for your careful check. The typo errors have been corrected one by one in the revised manuscript.
- Abbreviations should be explained in the text.
Response: All the abbreviations in the text has been examined and the full descriptions of the abbreviations have been supplemented in the revised manuscript when they first appeared.
- Authors wrote “absorption peak was mainly derived from the π–π transition”. Are they sure that there is π–π transition?
Response: According to B. Zhu, F. Wang, P. Li, C. Wang, Y. Gu, Surface oxygen-containing defects of graphene nanosheets with tunable nonlinear optical absorption and refraction, Phys. Chem. Chem. Phys. 20 (2018) 27105-27114, the conclusions are as follows.
In carbon materials containing a mixture of sp2 and sp3 bonding, the opto-electronic properties are determined by the π states of the sp2 sites. The π and π* electronic levels of the sp2 clusters lie within the band gap of σ and σ* states of the sp3 matrix and are strongly localized. GO (Graphene oxide) is an oxide of graphene, which is composed of carbon and other oxygen-containing functional groups. The main absorbance peak attributed to π – π* transitions of C=C in as-synthesized GO occurs around ~ 200 nm which red shifts to ~ 260 nm upon reduction. So the absorption peak of GO was at 228.96 nm, and absorption peak was mainly derived from the π–π transition.
- 6 should be corrected.
Response: In order to display the absorption peak more clearly, the abscissa of Fig. 6 has been shortened. In the revised manuscript, the abscissa wavelength of Figure 6 has been modified to the range of 200 nm to 400 nm. To state the absorption difference of the samples, we have calculated the bandgap, which is 3.38 eV, 3.16 eV, 3.25 eV, and 3.60 eV for T1, T2, T3, and T4, respectively.
- It is required to compare the nonlinear values ​​of the studied materials with similar ones.
Response: Thanks for your suggestion. We have compared the nonlinear values of our materials with that of graphene-based composites and added them in lines 256 to 261. Compared with the reported materials, the χ (3) of composite T2 (23.23 ×10-12 esu) was three times larger than that of graphene-γMnS (6.23 × 10-12 esu) [1], four times larger than that of α-MnS/rGO (4.93 × 10-12 esu) [2], six times larger than that of rGO-Au@CdS (3.43 × 10-12 esu) [3]. Meanwhile, χ (3) of composite T2 was 20 times larger than that of GeS2–Ga2S3–CDS chalcogenide glass (0.165 × 10-12 esu) [4], and six order of magnitude larger than that of CdS DQs (3.96×10-18 esu) [5]. Therefore, the ZrO2/RGO composite exhibited relatively strong NLO properties.
References:
(1) Z. Zhang, P. Li, P. Li, Y. Gu, Facile One-Step Synthesis and Enhanced Optical Nonlinearity of Graphene-gammaMnS, Nanomaterials (Basel) 9 (2019).
(2) Z. Zhang, B. Zhu, P. Li, P. Li, G. Wang, Y. Gu, Synthesis and third-order nonlinear optical properties of α-MnS and α-MnS/rGO composites, Optical Materials 92 (2019) 156-162.
(3) Y. Hao, L. Wang, B. Zhu, Y. Zhang, Y. Gu, Regulation and enhancement of the nonlinear optical properties of reduced graphene oxide through Au nanospheres and Au@CdS core-shells, Opt Express 29 (2021) 9454-9464
(4) Y. Hou, Q. Liu, H. Zhou, C. Gao, S. Qian, X. Zhao, Ultrafast non-resonant third-order optical nonlinearity of GeS2–Ga2S3–CdS chalcogenide glass, Solid State Communications 150 (2010) 875-878.
(5) A.C. Kuriakose, S. Udayan, T. Priya Rose, V.P.N. Nampoori, S. Thomas, Synergistic effects of CdS QDs – Neutral red dye hybrid system on its nonlinear optical properties, Optics & Laser Technology 142 (2021).
- Authors should read and cite the following article: Materials 13, 2559 (2020).
Response: We have cited these articles in the revised manuscript as Ref. [31].
- More interpretation of the results should be added.
Response: Thanks for your helpful suggestions. Experiment of femtosecond transient absorption is a useful tool to prove the charge transfer process in the composite samples. We have also done this work in our previous published articles to confirm the charge transfer process in composites (L. Li, F. Wang, Y. Liu, F. Cao, B. Zhu, Y. Gu, Local-Field-Dependent Nonlinear Optical Absorption of Black Phosphorus Nanoflakes Hybridized by Silver Nanoparticles, The Journal of Physical Chemistry C 125 (2021) 15448-15457). This paper is added in the revised manuscript as Ref. 36 to enhance persuasiveness. However, to deeply study the charge transfer between semiconductor particle and RGO, we are carrying out a series of time-resolved spectroscopy measurements and doing theoretical simulation by FDTD calculation. These results will be presented in our future articles in detail.

Round 2
Reviewer 2 Report
The authors revised the original version of the manuscript by referring to reviewer’s comments and questions. The revised manuscript appears publishable in the Nanomaterials but a further revision is recommended.
1, 2, and 5 Ok.
- Temperature (170 vs. 180 degree C) and pH differences (pH 8 vs. 9) the authors are arguing for the difference in the XRD patterns is too minor. It is unclear what the authors meant by the XRD standard card, so further explanation is needed. The reviewer strongly suggests that the authors add the discussion of the differences in the XRD patterns from the previous reports.
- The designation of the absorption maximum wavelengths for T1-T4 samples still seems inaccurate, especially between the samples of T1-T3. The reviewer strongly suggests the authors find the correct absorption maximum wavelengths of the samples for the accurate evaluation of the bandgaps. The background subtraction for the substrate absorption band at <200 nm with the long tails in the 200-400 nm may help.
Reviewer 3 Report
There is no π–π transition! It should be corrected!
These are for example: σ → σ* , π → π* , n → σ* , n → π*
The reference style should be checked.
Author Response
There is no π–π transition! It should be corrected!
These are for example: σ → σ* , π → π* , n → σ* , n → π*
The reference style should be checked.
Response:Thank you so much for your careful check. The π→π * in the revised manuscript has been corrected.